# Acoustic Aerosol Delivery: Assessing of Various Nasal Delivery Techniques and Medical Devices on Intrasinus Drug Deposition

**DOI:** 10.3390/ph16020135

**Published:** 2023-01-17

**Authors:** Lara Leclerc, Nathalie Prévôt, Sophie Hodin, Xavier Delavenne, Heribert Mentzel, Uwe Schuschnig, Jérémie Pourchez

**Affiliations:** 1Mines Saint-Etienne, Université Jean Monnet Saint-Etienne, INSERM, Sainbiose U1059, Centre CIS, F-42023 Saint-Etienne, France; 2Université Jean Monnet Saint-Étienne, Mines Saint-Etienne, INSERM, Sainbiose U1059, F-42023 Saint-Etienne, France; 3Nuclear Medicine Unit, CHU Saint-Etienne, F-42055 Saint-Etienne, France; 4PARI GmbH, D-82319 Starnberg, Germany; 5PARI Pharma GmbH, D-82166 Gräfelfing, Germany

**Keywords:** maxillary sinus, pulsating aerosol, nebulizer, drug deposition

## Abstract

This study aims to evaluate the impact of the nasal delivery technique and nebulizing technologies (using different frequencies of oscillating airflow) for acoustic aerosol targeting of maxillary sinuses. Sodium fluoride (chemical used as a marker), tobramycin (drug used as a marker) and ^99m^Tc-DTPA (radiolabel aerosol) were used to assess the intrasinus aerosol deposition on a nasal cast. Two commercial medical devices (PARI SINUS nebulizer and NL11SN ATOMISOR nebulizer) and various nasal delivery techniques (one or two nostrils connected to the aerosol inlet, the patient with the soft palate closed or open during the acoustic administration of the drug, the presence or not of flow resistance in the nostril opposite to the one allowing the aerosol to be administered) were evaluated. The closed soft palate condition showed a significant increase in drug deposition even though no significant difference in the rest of the nasal fossae was noticed. Our results clearly demonstrated a higher intrasinus aerosol deposition (by a factor 2–3; respectively 0.03 ± 0.007% vs. 0.003 ± 0.0002% in the right maxillary sinus and 0.027 ± 0.006% vs. 0.013 ± 0.004% in the left maxillary sinus) using the acoustic airflow generated by the PARI SINUS compared to the NL11SN ATOMISOR. The results clearly demonstrated that the optimal conditions for aerosol deposition in the maxillary sinuses were obtained with a closed soft palate. Thus, the choice of the nebulizing technology (and mainly the frequency of the pulsating aerosol generated) and also the recommendation of the best nasal delivery technique are key factors to improve intrasinus aerosol deposition.

## 1. Introduction

Chronic rhinosinusitis (CRS) is a common disease affecting 4.5–12% of North American and European populations. This chronic disease has an impact on patients’ quality of life and high indirect costs [1]. In this pathology, the maxillary sinuses, which are poorly ventilated cavities within the facial bones communicating with the nasal cavity via the maxillary ostium, are affected [2]. CRS is defined as the presence of two or more symptoms, such as nasal blockage, obstruction, congestion, or nasal discharge (anterior or posterior) [3]. It has been demonstrated that spraying drugs in the nasal fossae with a metered-dose pump spray is not an effective method for drug delivery to maxillary sinuses [4]. In contrast, nebulization of antibiotics may be an alternative treatment to improve local drug action while reducing systemic side effects. This method would both better target anatomical areas such as the middle meatus and increase drug delivery [5]. However, reaching maxillary sinuses remains a challenge for micrometer-sized airborne droplets introduced into the nasal cavity during aerosol therapy [6,7,8,9].

Numerous factors influence the intrasinus drug deposition, including the aerodynamic size of the droplets, the presence of an acoustic airflow, and finally the sinus anatomy features (i.e., volume of the sinus cavity, diameter and length of the maxillary ostium). A scientifically comprehensible mechanism of aerosol transport from the nasal cavity to the sinus cavity is based on the acoustic drug delivery approach. Indeed, an acoustic wave (i.e., by humming or through a medical aerosol device specifically developed for CRS treatment) can be superimposed on the airborne droplets to generate a so-called “pulsating”, “sonic”, or “acoustic” aerosol [10]. All things considered, the underlying mechanism of acoustic drug delivery as an innovative method to target the sinuses is now fairly well known. The physical principle is to increase the transfer of airborne droplets from the nasal cavity to the sinus cavity through the oscillatory exchange of air across the maxillary ostium according to the Helmholtz resonator theory (Figure 1). This oscillating airflow resulting from the acoustic wave allows a pressure gradient to be generated between the nasal fossae and the sinus cavity, which is a poorly ventilated area [11,12,13]. Thus, it has been previously demonstrated that the superposition of an acoustic wave on the airborne droplets produced by a nebulizer can allow, in some cases, a significant improvement of intrasinus drug deposition [14,15,16,17]. The application of this sonic aerosol therapy was also used in the case of other pathologies such as cystic fibrosis [18].

However, in order to maximize drug deposition, an “optimal” acoustic wave (in terms of amplitude but also in terms of frequency, which must correspond to the theoretical resonance frequency) must be applied to the patient according to its anatomical characteristics [5]. The Helmholtz resonator formula predicting this theoretical resonance frequency is indeed based on only a few elementary geometric parameters. This rather simple equation (Figure 1) tells us that knowing on the one hand the volume of the sphere (i.e., the maxillary sinus cavity in our case) and on the other hand the length and diameter of the tube (i.e., the maxillary ostium in our case), a theoretical resonance frequency can be calculated. However, by nature, this resonance frequency, which is the guarantee of an optimal deposit in the maxillary sinuses using acoustic drug delivery, is highly patient specific. Thus, a significant variation in these geometrical parameters within a population will inevitably lead to a wide range of different theoretical resonance frequencies. Indeed, it is obvious that the inter-individual anatomical variability existing between patients, in the volume of the maxillary sinus and the ostium geometry, automatically leads to large differences in the resulting theoretical resonance frequency. Furthermore, even for the same patient with given anatomical characteristics, this resonance frequency can also change during the evolution of the CRS pathology, because on the one hand inflammation and mucus secretion can decrease the diameter of the ostium, which is more or less obstructed, and on the other hand the volume of the cavity can also vary due to the fact that the sinus is more or less filled with fluid. In other words, this suggests that the use of a single fixed frequency of acoustic waves generated by a medical aerosol therapy device is always a compromise to find and can in no way be ideal and highly effective in real life for a large number of patients.

Faced with this difficulty of the highly patient-specific nature of acoustic drug delivery, strategies have been proposed to use variable acoustic frequencies that have yielded good results in terms of aerosol penetration for different sinus geometries [19,20]. These methods are based on the use of a frequency sweep that allows all possible resonance frequencies in a specific range to be reached. More recently, following the same idea, the use of a music signal superimposed on airborne particles showed interesting performance for aerosol intrasinus targeting [21]. However, none of these research strategies using benchtop prototypes with variable acoustic frequencies have resulted in a medical device available to patients. In this context, the choice of commercialized medical devices using acoustic aerosol appears to be associated with the best nasal delivery technique (e.g., one or two nostrils connected to the aerosol inlet, the patient with the soft palate closed or open during the acoustic administration of the aerosol, the presence or not of flow resistance in the nostril opposite to the one allowing the aerosol to be administered) as critical elements pending hypothetical new technological developments in the coming years using the propositions previously mentioned. Even if the literature is very scarce on this topic of performances associated with nasal delivery technique, different nebulization devices exist on the market. They are mainly jet nebulizers superimposed with an acoustic wave (each manufacturer choosing as a compromise a given acoustic frequency and amplitude of airflow for their devices) for better sinus targeting [8,12]. Different nasal delivery techniques are also recommended by aerosol medical device manufacturers with different positions of the soft palate during the nebulization process (closed or open) and various types of nasal interface (e.g., nasal joining piece in both nostrils, or nasal joining piece in one nostril and occlusion of the second nostril with a nose plug as a flow resistor). Recent studies described the use of the CFD method to measure the normative range of biomarkers in the human nasal cavity of adults and an in vitro study investigating nasal drug delivery in infants and children [22,23,24,25].

The aim of this study was to assess various nasal delivery techniques for two commercially acoustic aerosol devices with a similar mass median aerodynamic diameter (MMAD) following the manufacturer instructions: PARI SINUS nebulizer −3.2 µm MMAD, PARI GmbH, Starnberg, Germany, and NL11SN ATOMISOR nebulizer −4.1 µm MMAD, DTF Medical, Saint-Etienne, France. The deposition was evaluated using a nasal replica made by 3D printing showing two different ostium morphologies. The main objectives were to compare nasal delivery techniques and commercialized aerosol devices (using different frequencies and amplitudes of airflow) on the intrasinus drug deposition and the nasal cavity drug deposition.

## 2. Results

### 2.1. Aerosol Deposition Using NaF as a Tracer

The results of aerosol deposition using NaF as a tracer are summarized in Figure 2. Opened vs. closed soft palate conditions using the PARIS SINUS technology were compared, showing a significant increase in drug deposition in the left maxillary sinus (i.e., the short and broad ostium, Figure 2A), whereas no significant difference was noticed in the right maxillary sinus (i.e., the long and narrow ostium, see Figure 2A) or the rest of the nasal fossae (Figure 2B). Besides, compared to the opened soft palate conditions, the expelled fraction was higher and no thoracic fraction was observed for the closed soft palate position (Figure 2B).

### 2.2. Aerosol Deposition Using Tobramycin as a Tracer

The results of aerosol deposition using tobramycin as a tracer are summarized in Figure 3. The two commercial acoustic drug delivery devices (NL11SN vs. PARI SINUS) were compared. A significantly higher deposition was noticed in the different anatomy of maxillary sinuses for the PARI SINUS nebulizer compared to NL11SN (0.03 ± 0.007% vs. 0.003 ± 0.0002% in the RMS, respectively, and 0.027 ± 0.006% vs. 0.013 ± 0.004% in the LMS, respectively), with a non-significant difference for deposition in the rest of the nasal fossae. The thoracic fraction for the NL11SN (used in a very different delivery technique compared to the PARI SINUS: opened soft palate condition, aerosol administration occurring simultaneously in both nostrils without flow resistor and continuously) was similar to the expelled fraction for the PARI SINUS. Non-significant differences for the deposition in the nasal fossae were observed even through the NL11SN deposition seemed higher.

### 2.3. Aerosol Deposition Using ^99m^Tc-DTPA as a Radiotracer

The qualitative evaluation of the deposition was performed using ^99m^Tc-DTPA as a tracer by a qualified nuclear medicine specialist and the nuclear activity was quantified in each 3D ROI. The radioactivity was quantified in different regions of interest (Figure 4).

The results of the three individual experiments are displayed in Figure 5 and show that for the PARI SINUS the deposition pattern was:~0.05% in each maxillary sinus (0.048 ± 0.006% in the left maxillary sinus and 0.053 ± 0.013% in the right maxillary sinus);~28% of the emitted fraction in the nasal fossae (27.56 ± 0.01%);~23% of the emitted fraction in the interface (22.67 ± 1.76%), with the interface corresponding to the nostril joining piece and the plasticine used for sealing;~29% of the emitted fraction in the expelled fraction (29.67 ± 9.61%);~20% of the emitted fraction lost during the experiment (setup sealing).

**Figure 5 pharmaceuticals-16-00135-f005:**
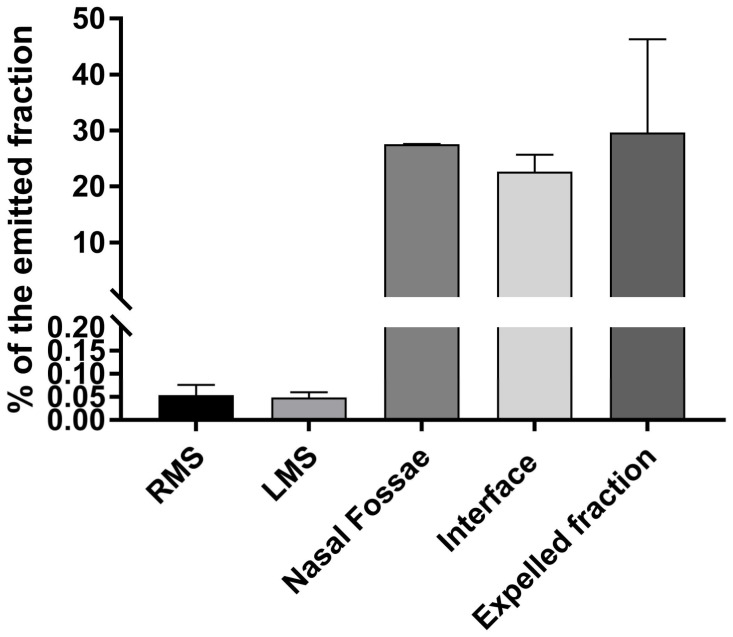
Results obtained after ^99m^Tc-DTPA nebulization with the PARI SINUS (closed soft palate condition). Results are expressed as percentage of the emitted fraction.

The calculation of an “activity balance” was performed for the PARI SINUS nebulizer with the respective deposited fraction in the sinus cavities, nasal fossae, and expelled aerosol. Approximately 80% of the total emitted aerosol was recorded in the planar scintigraphy acquisitions. The missing 20% was attributed to the environmental loss during the nebulization process and particularly during the change of nostril. Besides, the determination of residual activity in the nebulizer at the end of experiment led to calculating the emitted dose, corresponding to ~35% of the initial load introduced initially into the nebulizer tank.

## 3. Discussion

### 3.1. Methodological Overview

A comparison of the results of aerosol deposition obtained in the maxillary sinuses (expressed as a percentage of the emitted fraction) for the PARI SINUS closed soft palate condition showed approximatively 0.01–0.02% of NaF collected in each maxillary sinus, approximatively 0.02–0.03% of tobramycin in each maxillary sinus, and 0.05% of ^99m^Tc-DTPA in each maxillary sinus.

The results of the intrasinusal deposition were coherent comparing these three quantification modalities (NaF, tobramycin, and ^99m^Tc-DTPA). We demonstrated that NaF and tobramycin were efficient tracers compared to ^99m^Tc-DTPA, which is considered a gold standard in aerosol deposition studies. This observation is interesting to note because nuclear medicine experiments are not as easy to plan as NaF or tobramycin setups. Moreover, as ostium morphology was different between the right and left maxillary sinuses in our replica, we can highlight that these different deposition methodologies seem efficient for a large spectrum of patient that in real life have huge morphological differences at the level of the sinuses.

### 3.2. Device Comparison and Impact of the Nasal Delivery Technique

The two devices’ performances were compared using tobramycin as a tracer in conditions as close as possible to patient use. Intrasinusal and nasal cavity tobramycin deposition using PARI SINUS vs. NL11SN ATOMISOR in the ideal conditions of use as recommended by manufacturers were studied. For similar MMAD, our results clearly demonstrated a higher intrasinus drug deposition (by a factor of 2–3—0.03 ± 0.007% vs. 0.003 ± 0.0002% in the RMS, respectively, and 0.027 ± 0.006% vs. 0.013 ± 0.004% in the LMS, respectively) using the PARI SINUS device associated with a specific nasal delivery technique (i.e., closed soft palate, only one nostril connected to the aerosol inlet with a flow resistor in the opposite nostril) compared to the NL11SN ATOMISOR (i.e., open soft palate, two nostrils connected to the aerosol inlet).

Different hypotheses can explain this observation:-There is an impact from the aerosol acoustic delivery because delivery frequencies are different between the two aerosol medical devices. For the NL11SN, 100 Hz frequency sound vibrations were emitted automatically in alternating sequences of approximately 15 s. For the PARI SINUS the pulsation frequency was 44 Hz.-There is an impact from the nasal delivery technique. For the NL11SN ATOMISOR the aerosol administration was performed with a nasal joining piece and occurred simultaneously in both nostrils without a flow resistor and continuously with an opened soft palate, whereas for the PARI SINUS nebulizer the aerosol administration was performed in only one nostril, whereas the other nostril was occluded with a flow resistor and alternatively in each nostril with a nostril switch in the middle of the nebulization session with a closed soft palate recommended.-There is an impact of both aerosol acoustic delivery frequency and delivery technique.

Indeed, the intrasinus NaF deposition with open vs. closed soft palate for the PARI SINUS led to showing a great impact of the soft palate position for a given medical aerosol device. Our results demonstrate that a significantly higher intrasinus deposition was observed where the soft palate was closed in comparison to the opened soft palate condition (by a factor of at least 3 in the maxillary sinuses). The underlying mechanism of acoustic drug delivery as an innovative method to target the sinuses is to increase the transfer of airborne droplets from the nasal cavity to the sinus cavity by the oscillatory exchange of air across the maxillary ostium. This oscillating airflow resulting from the acoustic wave allows for the generation of a pressure gradient between the nasal cavity and the sinus cavity. Further research is needed (including CFD simulation) to better understand the impact of soft palate closure in this context, but a modification in the pressure gradient between the maxillary ostium and the nasal cavity could occur due to the change in soft palate position and thus contribute to improved acoustic aerosol delivery in maxillary sinuses. This could mainly explain the differences between the two devices. Benefits of closing soft palate have already been demonstrated [26], but patient compliance is a key factor in this case. Around 30 to 50% of patients do not take their treatments adequately regardless of the aerosol technology used, engendering many health costs [27]. This weak compliance is a probable cause of reduced efficacy resulting from the low percent of active pharmaceutical ingredients reaching the anatomical targets.

If we consider only the open soft palate condition, and experiments with the PARI SINUS and with the NL11SN ATOMISOR, aerosol deposition in the maxillary sinuses led to a similar amount (between 0.005% and 0.01%). These results seems to indicate that the aerosol acoustic delivery frequency is a minor factor compared to the soft palate position during the nebulization.

Moreover, we noticed that the tobramycin aerosol deposition in the nasal fossae and in the expelled fraction using the PARI SINUS allowed the same order of magnitude of aerosol deposition observed in the nasal fossae to be obtained using the NL11SN (i.e., ~10 wt%). Additionally, there was no deposition in the thoracic region when the soft palate was closed. These data are important for better targeting the aerosol regional deposition and, for example, to avoid side effects due to the deposition of drugs such as antibiotics in non-desired areas (such as the thoracic region in the case of sinus damage). Indeed, it is known that inhaled drugs are localized to the target organ, which generally allows for a lower dose than is necessary with systemic delivery (oral or injection), and thus fewer and less severe adverse effects [28,29].

This work is an experimental in vitro study that allowed us to highlight the major impact of various nasal delivery techniques and medical devices on intrasinus drug deposition. The main objective of this publication is therefore to propose experimental results, for example, on the impact of the position of the soft palate on the intrasinus deposit. These very original experimental results, which are proposed for the first time in the literature to our knowledge, will allow experts of CFD simulation and clinicians to continue this work in silico and in vivo in order to improve our knowledge on acoustic aerosol delivery in maxillary sinuses.

## 4. Material and Methods

### 4.1. Nasal Cast Replica

In this study, a nasal replica of a human plastinated cast was used (male specimen over 65 years old). The plastination technique, the methods of obtaining the human plastinated cast, its anatomic characteristics, and the advantages it provides to the aerosol deposition experiments were described in previous studies [15,16,17]. The technique used to segment and reconstruct the nasal airways from computer tomography (CT) images was fully described in 2014 [17]. Briefly, CT scan images of the human plastinated cast were numerically analyzed to obtain the dimensions of ostia and the volume of maxillary sinuses (MS). A nasal, transparent, water-resistant, non-porous resin replica of the human plastinated cast, created using a stereolithography technique, was used to perform the deposition experiments. Anatomical and aerodynamic reproducibility between the nasal replica and its human plastinated cast were confirmed after performing endoscopy and CT scans. The reliability of the nasal cavity geometry (determined by acoustic rhinometry) and airflow resistance (measured by rhinomanometry) of the nasal replica were previously described [30]. Rhinomanometry was used to provide a quantification of nasal airway resistance. From the pressure vs. flow curves, the unilateral airflow resistances found for the left and right nasal cavities were perfectly similar at 0.18 Pa.s.cm^−3^ (i.e., 1.8 cmH_2_O.s.L^−1^). The bilateral airflow resistance was logically measured at a lower value compared to unilateral resistances, around 0.13 Pa.s.cm^−3^ [30]. The geometry of the nasal cavities was also characterized using acoustic rhinometry. We always found a minimal cross-sectional area of around 0.5 cm^2^ and a cross-sectional area higher than 1.5 cm^2^ from the middle meatus region [30].

As previously characterized, the left maxillary ostium appears to be short and broad, whereas the right maxillary ostium is long and narrow [15,16,17]. This remarkable sinus anatomy variation on the same nasal cast allows the impact of the ostium geometry on drug delivery to be assessed by comparing left to right intrasinus deposition.

### 4.2. Aerosol Deposition Using NaF as a Tracer

Intrasinus deposition of the PARI SINUS (PARI GmbH, Starnberg, Germany) nebulizer was evaluated using the nasal replica and sodium fluoride (NaF) as a tracer in two experimental conditions:-Closed soft palate (n = 6); no ventilation. The nasal cast trachea was occluded with a sealing cap.-Open soft palate (n = 6); ventilation of the nasal replica with the PARI Compas 2 pump (I/E = 1/1, Vt = 500 mL, 15 breath/min).

The flow rate value generated by the PARI SINUS compressor was 4.6 L/min.

During the experiment, the nebulizer was tightly connected to only one nostril, whereas the other nostril was connected to a commercial nasal plug (acting as a flow resistor) with a PARI filter 41B0523 (PulmoMed, Nanterre, France) to calculate the expelled fraction. The PARI SINUS nebulizer was always used with the PARI SINUS compressor (PARI GmbH, Starnberg, Germany).

The aerosol administration was performed with a fixed nebulization duration (i.e., 3 min of aerosol administration by the left nostril, then 3 min of aerosol administration by the right nostril, whereas the other nostril was occluded with a flow resistor as the procedure recommended by the manufacturer for patient use). For all experiments, the volume of NaF in the nebulizer was fixed at 4 mL. Illustrations of the setup for both conditions (closed and opened soft palate) are shown in Figure 6 and Figure 7.

NaF 2.5% (M/V) was used as a chemical marker according to European standard procedure (NF EN 13544–1). The maxillary sinuses of the nasal replica were hermetically sealed during the experiments. At the end of each nebulization, the plates sealing the sinuses were then removed and the maxillary sinuses were flushed with a syringe containing 1 mL of distilled water. Each sinus was flushed a couple of times using the same distilled water. The region close to the maxillary ostium was never flushed. Besides, the nasal cavity of the model and filters (PARI filter 41B0523, PulmoMed, Nanterre, France) to assess the expelled fraction and the thoracic fraction (only for the opened soft palate condition) were washed with 8 mL of distilled water each. The thoracic fraction corresponds to “the fraction of inhaled particles penetrating beyond the larynx” [31].

The concentration of deposited NaF aerosol into maxillary sinuses, the nasal cavity, and the expelled and thoracic fraction was measured using the perfect Ion Fluoride Electrode (Mettler Toledo, Greifensee, Switzerland) and the ion meter (Mettler Toledo, Greifensee, Switzerland) after adding 5% of TISAB IV solution (Sigma-Aldrich, Darmstadt, Germany) to every rinse liquid, according to the European standard procedure (NF EN 13544–1). The NaF concentration remaining in the nebulizer at the end of the experiment was also assessed.

After every rinsing procedure, the nasal replica was removed from the setup and copiously washed with distilled water and dried with compressed air. To verify the efficiency of model washing, distilled water nebulization experiments were randomly performed throughout the experiments by filling the nebulizers with 4 mL of distilled water instead of NaF.

### 4.3. Aerosol Deposition Using Tobramycin as a Tracer

The deposition of the PARI SINUS nebulizer (PARI GmbH, Starnberg, Germany) was compared to the deposition of the NL11SN ATOMISOR (DTF Medical, Saint-Etienne, France) in the nasal replica using tobramycin (Vantobra, PARI Pharma GmbH, Gräfelfing, Germany, 170 mg in 1.7 mL; 100 mg/mL) as a tracer in different conditions in order to strictly follow the manufacturer recommendations for patient use:-PARI SINUS: Experiments were performed with the nasal replica with a closed soft palate (n = 6). During the experiment, the nebulizer was tightly connected to one nostril while the other nostril was connected to PARI filter 41B0523 (PulmoMed, Nanterre, France) with a flow resistor to calculate the expelled fraction. The PARI SINUS nebulizer was always used with the recommended PARI SINUS compressor. The aerosol administration was performed in only one nostril at a time with a fixed nebulization duration (i.e., 3 min in the left nostril and 3 min in the right nostril). During the experiment, the other nostril was occluded with a flow resistor as the procedure recommended by the manufacturer for patient use. For all experiments, the volume of NaF in the nebulizer was fixed (4 mL). In this case the aerosol administration occurred alternatively in each nostril, with a nostril switch in the middle of the nebulization session. The illustration of the setup is the same as that shown in Figure 6.

The flow rate value generated by the PARI SINUS compressor was 4.6 L/min.

-NL11SN ATOMISOR: The AOHbox compressor (DTF Medical, Saint-Etienne, France) was used and a nebulization duration was fixed to 10 min with the soft palate open (n = 6). The nasal replica ventilated with the PARI Compas 2 pump (I/E = 1/1, Vt = 500 mL, 15 breath/min) and was connected to the pump with a 30 cm-long tube simulating the trachea. The aerosol administration was performed with a nasal joining piece according to the procedure recommended by the DTF manufacturer for patients. In this case aerosol administration occurred simultaneously in both nostrils without a flow resistor and continuously. The thoracic fraction was collected on a PARI filter 41B0523 (PulmoMed, Nanterre, France) localized between the nasal cast and the respiratory pump. The volume of tobramycin in the nebulizer was fixed (4 mL). Illustrations of the setup are shown in Figure 8.

The flow rate value generated by the ATOMISOR compressor was 6.5 L/min.

The procedure involved filling the nebulizers with 4 mL of a tobramycin solution as a marker. At the end of each nebulization, the plates sealing the sinuses were then removed and the maxillary sinuses were flushed with a syringe containing 0.5 mL of distilled water. Each sinus was flushed a couple of times using the same distilled water.

The region close to the maxillary ostium was never flushed. Besides, the nasal cavity of the model and filters to assess the expelled fraction and the thoracic fraction (only for the opened soft palate condition) were washed with 8 mL of distilled water each. The tobramycin concentration remaining in the nebulizer at the end of the experiment was also assessed.

The tobramycin concentrations in the samples were quantified by liquid chromatography tandem mass spectrometry using the Acquity UPLC system coupled with a Xevo TQD triple quadrupole (Waters, Saint-Quentin-en-Yvelines, France). The samples were diluted with internal standard and acetonitrile. After centrifugation, 5 µL of supernatant were injected. An elution gradient was applied for 2.5 min to a kinetex HILIC column (30 mm × 3 mm; 2.6 µm) (Phenomenex, Le Pecq, France). The transitions monitored were mass/charge (*m*/*z*) 468.24→205.13 and 484.28→324.27 for tobramycin and kanamycin B internal standard.

### 4.4. Aerosol Deposition Using ^99m^Tc-DTPA as a Radiotracer

PARI SINUS nebulizer deposition in the nasal replica using ^99m^Tc-DTPA as a radiotracer was evaluated with single-photon emission computed tomography (SPECT) imaging. Experiments were performed on the nasal replica with a closed soft palate (n = 3). The aerosol administration was performed in only one nostril with a fixed nebulization duration (i.e., 3 min in the left nostril and 3 min in the right nostril). During the experiment, the nebulizer was tightly connected to one nostril while the other nostril was connected to a PALL filter BB50TE (with a flow resistor) to calculate the expelled fraction. The volume of ^99m^Tc-DTPA tracer in the nebulizer was fixed (4 mL). The PARI SINUS nebulizer was always used with the recommended PARI SINUS compressor and three individual experiments were performed. Illustrations of the setup are shown in Figure 9.

SPECT generated a 3D description of a gamma-emitting radionuclide distribution using the data obtained from a rotating gamma camera. SPECT images were improved by the addition of anatomical image data. To increase the accuracy of the radioactivity distribution, the counts detected by the gamma camera were corrected for tissue attenuation, improving the quality of quantification. SPECT and CT acquisitions were performed with a SYMBIA T2 variable angle dual detector SPECT with a two-slice spiral CT for rapid, accurate attenuation correction and anatomical mapping (Siemens, Knoxville, TN, USA).

Nebulizers were filled with 100 mega Becquerel (MBq) of DTPA-Technetium 99 m (^99m^Tc-DTPA) solution (DTPA Technecsan^®^, Covidien-Mallinckrodt, ND, USA) in a 4 mL syringe. Immediately after aerosol administration, each element was successively set up on the examining table in the center of the gamma camera field. Planar images were obtained for the full and empty syringe with a needle (60 s anterior/posterior), the nebulizer, the interface (nasal joining piece and plasticine), the expelled fraction filter (filter with flow resistor), and the nasal replica with maxillary sinus sealing plates called the “head” (180-s anterior/posterior).

The background noise on the planar images (256 × 256 image matrix, zoom 1) was recorded and suitable corrections were made to take into account the radioactive decay and the attenuation of gamma rays by the different setup elements. Images obtained for the syringe and the nebulizer, and a supplementary image acquisition of all the rest of the material in possible contact with radionuclides, were used to evaluate the quality control of the nebulization experiment. For each experiment an activity balance was verified between the total radioactivity deposited and the total radioactivity nebulized.

Without moving the plastinated nasal cast, a 3D SPECT acquisition was performed with 64 projections (2 × 32) of images of 30 s each. Finally, a CT scan was performed with the following parameters: 130 kV, 90 mAs, slice thickness 1.25 mm, increment 0.9 mm, pitch 1.6, rotation time 1.5 s.

A multimodality computer platform (Symbia net; Siemens Healthcare SAS, Saint-Denis, France) was used for image review and manipulation. Both the transmission and emission scans were reconstructed using 3D OSEM (8 subsets, 5 iterations), with a pre-reconstruction smoothing using a 3D Butterworth filter (cutoff: 0.45 cycles/cm; order 5), 128 × 128 image matrix, zoom 1.23, and pixel size 3.9 mm. SPECT images were reconstructed using scatter correction (scatter energy window) and CT attenuation correction. CT and SPECT images were matched and fused into transaxial images.

Different 3D regions of interest (ROI) were manually drawn on the CT images with the software: right maxillary sinus, left maxillary sinus, central nasal cavity, and global head. Frontal sinuses were not analyzed because no radioactivity was found in this region. Ethmoid sinuses were impossible to distinguish from the central nasal cavity due to the Compton effect, so analyses in this region were not accurate. Quantification in the maxillary sinuses was performed on the hot spots observed and not with the global isocontour due to Compton effect with the central nasal cavity.

## 5. Conclusions

Aerosol intrasinus deposition with open vs. closed soft palate shows a great impact on the soft palate position for a given aerosol device. Thus, the impact of the aerosol delivery technique (and, in particular, the soft palate position), in addition to the technological impact of the aerosol device (and, in particular, the frequency and amplitude of the acoustic aerosol), is a key factor to improve aerosol deposition to maxillary sinuses.

## Figures and Tables

**Figure 1 pharmaceuticals-16-00135-f001:**
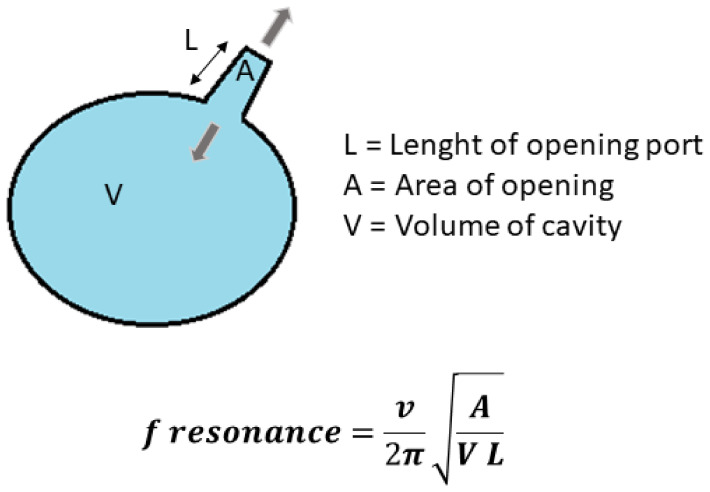
Helmholtz resonator formula. Where *v* is the Sound Speed in Air, and *A*, *L* and *V* Respectively the Section and the Length of the Tube, and the Volume of the Cavity.

**Figure 2 pharmaceuticals-16-00135-f002:**
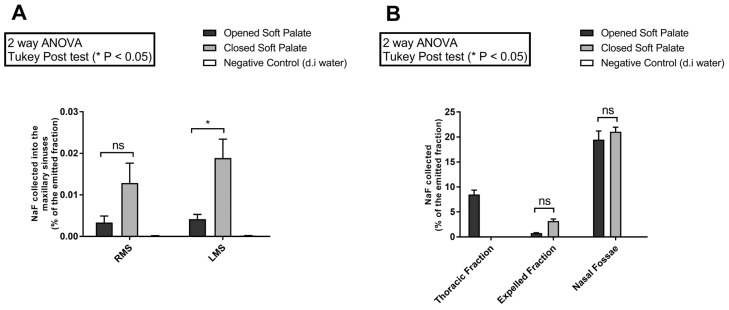
Comparison of results obtained for the NaF deposition with opened or closed soft palate for the PARI SINUS device. (**A**) Maxillary sinus deposition (RMS = right maxillary sinus; LMS = left maxillary sinus). (**B**) Amount of NaF collected in the thoracic fraction (only for opened soft palate condition), in the expelled fraction, and in the nasal fossae. Results are expressed in percentage of the emitted fraction. Negative control corresponds to deionized water (d.i water) nebulization. Statistically different from control cells: * *p* < 0.05.

**Figure 3 pharmaceuticals-16-00135-f003:**
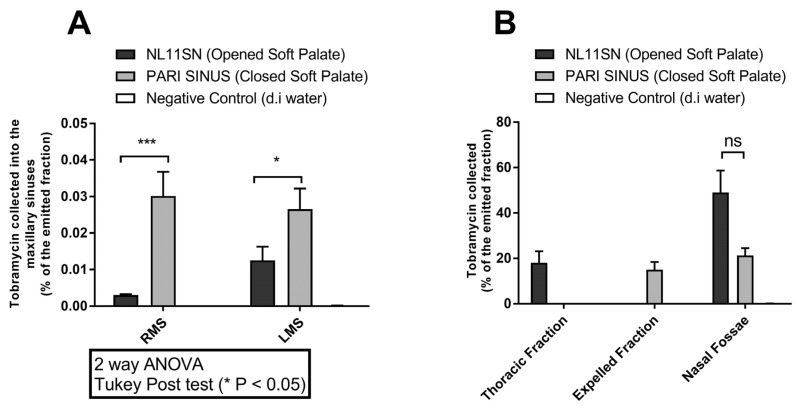
Comparison of results obtained for the tobramycin deposition with the NL11SN device and the PARI SINUS device. (**A**) Maxillary sinus deposition (RMS = right maxillary sinus; LMS = left maxillary sinus). (**B**) Amount of tobramycin collected in the thoracic fraction (only for opened soft palate condition), in the expelled fraction, and in the nasal fossae. Results are expressed in percentage of the emitted fraction. Negative control corresponds to deionized water nebulization. Statistically different from control cells: *** *p* < 0.001, * *p* < 0.05.

**Figure 4 pharmaceuticals-16-00135-f004:**
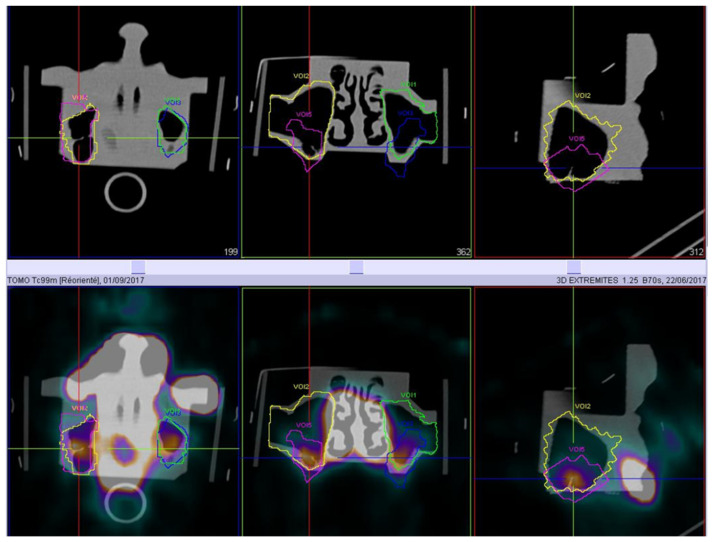
Maxillary sinuse ROI delimitations for ^99m^Tc-DTPA nebulization with the PARI SINUS device. The illustration shows the SPECT images in transverse, coronal, and sagittal views with deposition of aerosol in the upper airways. Yellow and green isocontours correspond to the total maxillary sinus cavity; pink and blue isocontours correspond to hot spot delimitation in the maxillary sinuses (other hot spots correspond to deposition on the external side of the human nasal cast).

**Figure 6 pharmaceuticals-16-00135-f006:**
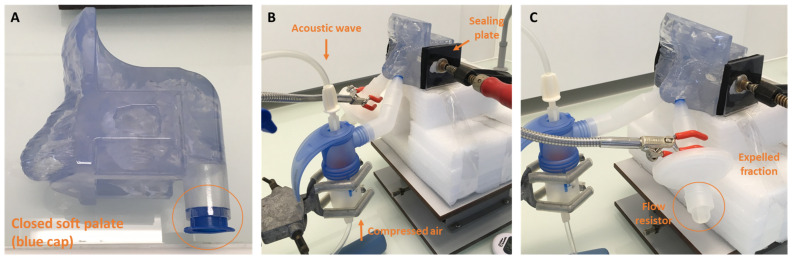
Experimental setup for NaF nebulization with PARI SINUS device in the closed soft palate condition. (**A**) Soft palate closed with a blue sealing cap. (**B**) The illustration shows the human nasal cast with closed soft palate and sealing of maxillary sinuses. The nebulization duration was fixed at 6 min—3 min of aerosol administration by the left nostril (**B**), then 3 min of aerosol administration by the right nostril (**C**)—whereas the other nostril was occluded with a flow resistor as the procedure recommended by the manufacturer for patient use.

**Figure 7 pharmaceuticals-16-00135-f007:**
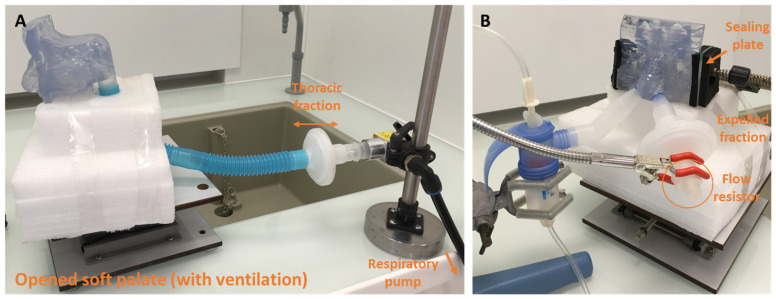
Experimental setup for NaF nebulization with PARI SINUS device in the opened soft palate condition. (**A**) The illustration shows the human nasal cast with open soft palate connected to the respiratory pump. The thoracic fraction was collected on a PARI filter 41B0523. (**B**) The nebulization duration was fixed at 6 min with sealing of maxillary sinuses (3 min in each nostril, whereas the other nostril was occluded with a flow resistor as the procedure recommended by the manufacturer for patient use). The expelled fraction was collected on a PARI filter 41B0523.

**Figure 8 pharmaceuticals-16-00135-f008:**
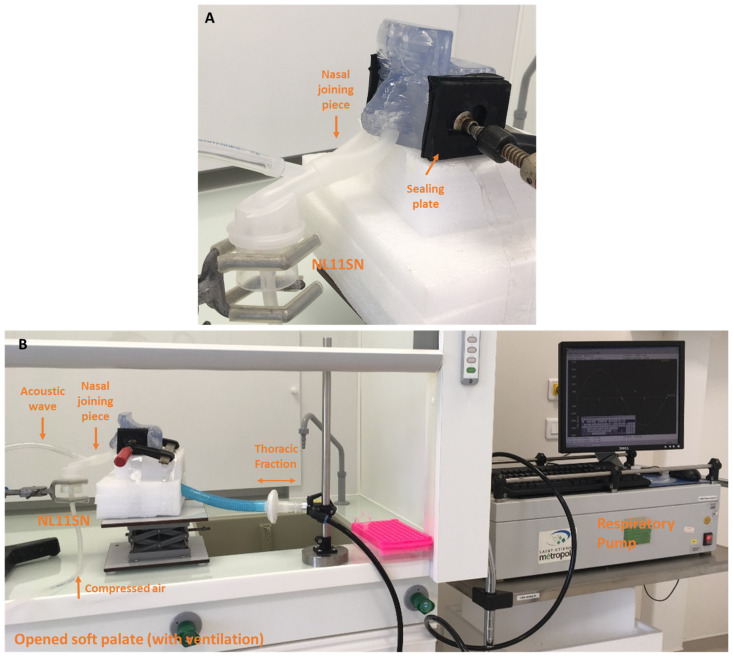
Experimental setup in opened soft palate condition for tobramycin nebulization with the NL11SN device. (**A**) Illustration showing the human nasal cast assembly with sealing of maxillary sinuses and nasal joining piece. (**B**) Illustration showing the human nasal cast assembly with open soft palate connected to the respiratory pump. The thoracic fraction was collected on a PARI filter 41B0523. The nebulization duration was fixed at 10 min with the nasal joining piece.

**Figure 9 pharmaceuticals-16-00135-f009:**
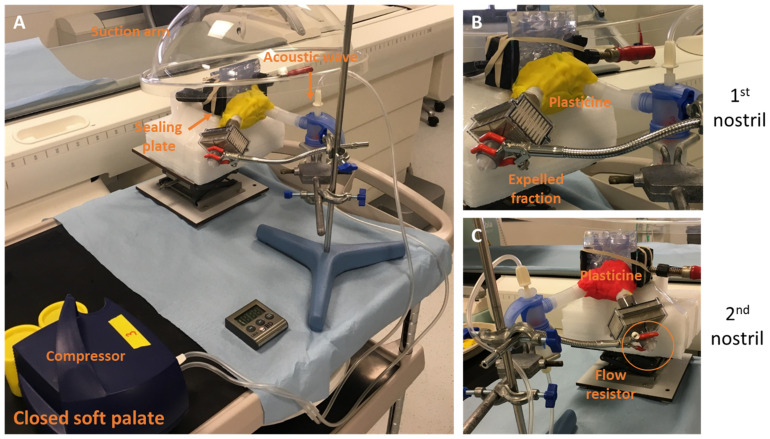
Experimental setup in the closed soft palate condition for ^99m^Tc-DTPA nebulization with the PARI SINUS device in the Nuclear Medicine Department under a suction arm. (**A**) The illustration shows the human replica assembly with closed soft palate and sealing of maxillary sinuses. (**B**,**C**) PALL Filter BB50TE with a flow resistor was used to collect the expelled fraction. The nebulization duration was fixed at 6 min (3 min in each nostril). Improved sealing was obtained using plasticine to avoid radioactivity leakage.

## Data Availability

Data is contained within the article.

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
