# Peer review of "Acoustic Aerosol Delivery: Assessing of Various Nasal Delivery Techniques and Medical Devices on Intrasinus Drug Deposition"

_pharmaceuticals, 2023, doi:10.3390/ph16020135_

Round 1
Reviewer 1 Report
The article is well presented and deals with a topic not yet thoroughly investigated in the literature. The set-up of the experiments is very robust and I have no major criticisms except some points that I suggest are implemented:
in the introduction I would also mention the application of this therapy for cystic fibrosis
specify where you collected the "thoracic fraction". This is not a conventional definition.
enter the flow-rate value generated by the compressor that reaches the ampoule
give an explanation to the highest deposition where the soft palate is closed
in the conclusions and in the abstract I suggest to insert some numerical value obtained from the results
Reviewer 2 Report
This is a good in vitro study of intra-nasal drug delivery assessing different inhaled medical devices.
The manuscript is useful to provide researchers information of experimental study on aerosol deposition in nasal cavity. However, there are still many questions need to be addressed before publication. Below are specific major comments:
1. In Introduction section: Refer to the recent study, which presents CFD method used to measure normative range of biomarkers in human nasal cavity of adults and in vitro study investigating nasal drug delivery in infants and children:
- Borojeni et al. (2020), “Normative ranges of nasal airflow variables in healthy adults”,
International Journal of Computer Assisted Radiology and Surgery, volume 15, pages 87–98 (2020)
- Amirav et al. (2015), “Nasal Versus Oral Aerosol Delivery to the “Lungs” in Infants and Toddlers”, Journal of Pediatric Pulmonology, Vol. 50, 2015, pp:276-283
- Janjian Wei et al. (2016), “Low re-inhalation of the exhaled flow during normal nasal breathing in a pediatric airway replica”, Journal of Building and Environment, Vol. 97, 2016, pp:40-47.
- Garcia et al. (2009), Interindividual variability in nasal filtration as a function of nasal cavity geometry. Journal of Aerosol Medicine and Pulmonary Drug Delivery, Volume 22, Number 2, 2009
2. Please provide the technique used to segment and reconstruct the nasal airways from CT images.
And refer to the following publications:
- Borojeni et al. (2016), “Creation of an idealized nasopharynx geometry for accurate computational fluid dynamics simulations of nasal airflow in patient-specific models lacking the nasopharynx anatomy” International Journal for Numerical Methods in Biomedical Engineering, 2016.
- Borojeni et al. (2014), “Measurements of total aerosol deposition in intrathoracic conducting airway replicas of children”, Journal of Aerosol Science, Vol. 73, 2014, pp:39-47
3. Provide the subject information such as age, gender, weight, and height in the manuscript.
4. Provide volume and surface area of each nasal airway/replica.
5. Please validate your experimental results with previously publishes data obtained from numerical CFD study/in vitro/in vivo study.
6. Please provide intranasal pressure loss distribution in each replica for different inhalation flow rate.
7. Please provide a figure/plot showing nasal deposition efficiency vs. impaction parameter (da2Q) for 2 conditions, (A) once particles inhaled from right nostril of the nasal replica and (B) once particles inhaled from left nostril of the nasal replica.
8. Please measure and report nasal resistance for each nasal replica and provide it as a table in the manuscript.
9. Please provide geometric measurements (such as minimum cross-sectional area, Hydraulic diameter) of the nasal airway used in your study in the manuscript.
10. How did the authors validate their experimental results? It is necessary to compare your empirical results obtained from the experiment with the CFD results from literature.
Round 2
Reviewer 2 Report
Dear Authors,
Thank you very much for considering and applying reviewer comments to your manuscript.
This is an aerosol drug delivery study, so it is important to provide some information about the size of particles (particle diameter) released to each replica.
Furthermore, providing a plot showing nasal deposition efficiency vs. impaction parameter (da2Q) help your audience to validate their results with yours.
Please consider these 2 suggestions in your future research.
Best regards,